# Combinations of Antimicrobial Polymers with Nanomaterials and Bioactives to Improve Biocidal Therapies

**DOI:** 10.3390/polym11111789

**Published:** 2019-11-01

**Authors:** Roberto Yañez-Macías, Alexandra Muñoz-Bonilla, Marco A. De Jesús-Tellez, Hortensia Maldonado-Textle, Carlos Guerrero-Sánchez, Ulrich S. Schubert, Ramiro Guerrero-Santos

**Affiliations:** 1Centro de Investigación en Química Aplicada (CIQA), Boulevard Enrique Reyna No. 140, 25294 Saltillo, Mexico; hortensia.maldonado@ciqa.edu.mx; 2Instituto de Ciencia y Tecnología de Polímeros (ICTP-CSIC), C/Juan de la Cierva 3, 28006 Madrid, Spain; sbonilla@ictp.csic.es; 3Centro de Investigación y de Estudios Avanzados (CINVESTAV) Unidad Mérida, A.P. 73, Cordemex, 97310 Mérida, México; iqi.marco@gmail.com; 4Laboratory of Organic and Macromolecular Chemistry (IOMC), Friedrich Schiller University Jena, Humboldtstrasse 10, D-07743 Jena, Germany; carlos.guerrero.sanchez@uni-jena.de (C.G.-S.); ulrich.schubert@uni-jena.de (U.S.S.); 5Jena Center for Soft Matter (JCSM), Friedrich Schiller University Jena, Philosophenweg 7, D-07743 Jena, Germany

**Keywords:** biocide polymers, antibacterials, cationic copolymers

## Abstract

The rise of antibiotic-resistant microorganisms has become a critical issue in recent years and has promoted substantial research efforts directed to the development of more effective antimicrobial therapies utilizing different bactericidal mechanisms to neutralize infectious diseases. Modern approaches employ at least two mixed bioactive agents to enhance bactericidal effects. However, the combinations of drugs may not always show a synergistic effect, and further, could also produce adverse effects or stimulate negative outcomes. Therefore, investigations providing insights into the effective utilization of combinations of biocidal agents are of great interest. Sometimes, combination therapy is needed to avoid resistance development in difficult-to-treat infections or biofilm-associated infections treated with common biocides. Thus, this contribution reviews the literature reports discussing the usage of antimicrobial polymers along with nanomaterials or other inhibitors for the development of more potent biocidal therapies.

## 1. Introduction

The discovery of penicillin and other antibiotics in the mid-twentieth century promised to radically overcome the bleak scene of infectious diseases that had always plagued humankind. The notable effectivity of antibiotics led to an over-confidence that contagious diseases would be eradicated. However, antimicrobial resistance (AMR) continues to evolve, representing a serious concern worldwide. According to a recent report [1], 700,000 people die every year worldwide due to the drug resistance of common bacterial strains and it is expected that this number may rise to 10 million by 2050 with the corresponding undesirable impact on the world’s economy. This issue, is in part, attributed to the declining effectiveness of antibiotics against different microorganisms, which continuously evolve towards a stronger species as a result of prolonged overuse and misuse in human therapeutics [2,3,4]. The most essential mechanisms of antimicrobial resistance include the enzymatic degradation of antibiotic molecules, changes in membrane permeability to antibiotics and alteration of bacterial proteins that are antimicrobial targets [5]. In the last decades, new and novel antimicrobial agents, such as metallic nanoparticles, macromolecules with biocidal properties, carbon-based materials, reactive-oxygen-species or antimicrobial peptides, have been disclosed as a response to this complicated epidemiological situation. The aforementioned antimicrobials are able to kill microorganisms as bacteria and parasites through different mechanisms of action that include the disruption of cell walls in bacteria, and the delivery of metal ions and toxic radicals that inhibit oxidative enzymes or interfere with DNA/RNA replication, among others. Although these agents have biocidal activity in the treatment of bacterial infections when separately administered, it has been suggested that the co-administration of two or more mechanistically different antimicrobial agents can effectively fight bacterial infections by lowering the prescribed dose, preventing antimicrobial resistance and increasing the overall treatment efficacy [6,7,8,9]. The valuable contributions, such as those provided by Tängden [10] and Bassetti [11], discussed in detail the application of drug combination therapy for severe infections with multidrug-resistant Gram-negative bacteria. On the other hand, several approaches combine different therapies, such as the inhibition of targets via different pathways, which was highlighted by Fischbach and coworkers [7] and recently, Wright described a strategy for the development of effective antibiotic combinations [12]. In the field of materials, Allahverdiyev and coworkers addressed the interactions, mechanisms of action and synergistic effects of different metallic nanoparticle systems (i.e., Ag, Au, ZnO, and TiO_2_) acting in combination with some antibiotics (i.e., amoxicilin, penicilin) [13].

Inspired by these meaningful findings, this paper focuses on the investigations devoted to antimicrobial polymers combined or co-administrated with other biocides to enhance their effectiveness. To the best of the authors’ knowledge, this paper is the first to systematically review multicomponent systems containing at least one antimicrobial polymer (v.gr, cationic polymers, polymeric *N*-halamines, etc.) showing synergistic activity and their use for therapeutic purposes (Figure 1), as in the last two decades, antimicrobial polymers have emerged as a great alternative to conventional biocidal agents or antibiotics. Apart from their biocidal properties in solution, antimicrobial polymers have also contributed to the development of coatings, fibers and implants to prevent microbial contamination in a wide range of products [14,15].

## 2. Co-Administration of Two Different Antimicrobial Polymers

During recent decades, many research efforts have been directed to the development of novel polymers with intrinsic antimicrobial activity as alternative systems to conventional biocides and antibiotics. This movement is mainly stimulated by its high biocidal activity, low potential for building up antibiotic-resistance and reduced toxicity [16,17]. In particular, antimicrobial polymers have increasingly gained importance in the development of antimicrobial surfaces, which exert a biocidal effect without releasing any poisonous substance. Some contributions have reviewed several investigations describing the different types of antimicrobial polymers as well as their mechanism of action, the parameters related to their antimicrobial activity dosage amounts and their specific applications [15,18,19,20,21].

As an example, most prominent biocidal polymers are often those derived from (meth)acrylates, in particular, the ones containing quaternary ammonium salts (*v.gr*. quaternary phosphonium, guanidine, *N*-halamines, etc.). In this context, quite significant efforts have been focused on the design of polycationic systems, which are readily affordable from synthetic and economic points of view. In general terms, cationic polymers are highly efficient in fighting Gram-positive and gram-negative bacteria by ensuring electrostatic interactions due to their negatively charged cell wall. This interaction is a precondition for cell wall rupture or cell lysis. However, when polycations are used as surface coatings, their biocidal activity can significantly decrease due to the lack of mobility, i.e., polymer chains are highly confined and therefore unable to interact [22]. Some systems based on two different antimicrobial polymers or polymers with various biocidal functionalities have been disclosed to overcome this limitation. This strategy opens the opportunity to overcome potential limitations associated with each biocidal agent or related mechanism as well as the possibility to enhance their biocidal activity by promoting a synergistic effect. Thus, different polymeric systems containing a dual antimicrobial function are discussed in the following. 

### 2.1. Improving Antimicrobial Activity of Chitosan by Grafting Biocidal Polymers 

Chitosan (CS) is a naturally occurring polymer obtained through the alkaline deacetylation of chitin possessing many attracting properties including antimicrobial activity and nontoxicity [23]. Further, CS has shown antimicrobial activity against a wide range of microorganisms, including bacteria, fungi and yeasts [23,24,25]. A widely accepted mechanism of biocidal action for this polymer involves the protonation of its amino groups at the acidic pH value (lower than p*K*a) that lead to the formation of cationic groups, which electrostatically interact with bacterial cell walls. Thus, at the acidic pH value, the antimicrobial activity of CS is attributed to the electrostatic interactions and chelation effects. Nevertheless, CS can also be modified to enhance its solubility and to improve its biocidal properties. For instance, Hu et al. [26] investigated the antimicrobial activity of a dual system based on CS modified with guanidine derivatives. It is noted that guanidinium salts have also been investigated as medical and crop protection agents and antiseptics for industrial products due to their high antimicrobial and antifungal activity [27]. Research results highlighted that guanidinylated CS had a higher antibacterial activity, whose MIC in hydrochloric acid aqueous solution (pH 5.4) was four times lower than the one observed for pristine CS. Moreover, guanidinylated CS inhibited the growth of *S. aureus* and *B. subtilis* at pH 6.6. This enhanced capability of modified CS was attributed to an overall increase in the positive charge density as a consequence of the incorporation of guanidine derivatives, which promotes the adsorption of these polycations onto the negatively charged cells surface. Nevertheless, the quaternization of CS with alkylating agents is probably the most frequently employed approach to confer permanent positive charges and to improve biocidal activity [28]. Another common strategy is also the grafting of acidic derivatives, such as gallic and caffeic acids, onto CS via carbodiimide based coupling, free radical mediated grafting, among other synthetic approaches [29]. On the whole, most of the modified CS exhibits higher antimicrobial activity against Gram-positive and Gram-negative as compared to pristine CS [30,31]. This could be ascribed to an increase in the permeability of these polymers through the outer membrane in the case of Gram-negative bacteria and the cell membrane in Gram-positive bacteria producing the release of the intracellular components [32]. 

Aside from the preparation of CS with various biocidal functions, CS has also been combined with other antimicrobial polymers via the grafting or blending processes. For instance, poly[2-(acryloyloxy)ethyl]trimethylammonium chloride, was grafted onto CS chains to improve its antimicrobial activity in coatings as tested against Gram-positive *Staphylococcus aureus*, Gram-negative bacteria *Klebsiella pneumoniae* and *Aspergillus fumigatus* fungi [33]. Recent works mentioned the functionalization of chitosan with *N*-halamines [34] and pyridinium compounds [35] grafted as side groups with respect to the chitosan main chain in order to enhance the biocidal activity, taking advantage of the antimicrobial features of each one. Moreover, low molar mass antimicrobial polymer, e.g., *ε*-poly-l-lysine (EPL), have also been grafted onto CS chains [36], to improve antimicrobial activity against Gram-negative bacteria (*Escherichia coli* and *Pseudomonas aeruginosa*), Gram-positive bacteria (*Enterococcus faecalis* and methicillin-resistant *S. aureus* (MRSA)) and fungi (*Candida albicans* and *Fusarium solani*). Although these latter grafted copolymers revealed lower activity than pristine EPL against bacteria, they exhibited superior antifungal activity. This effect suggests the occurrence of a synergistic effect. Hou et al. recently disclosed an additional peptide-CS based system: [37] Self-assembled cationic nanoparticles derived from CS-graft-oligolysine (CSM5-K5) were combined with CS. This formula showed a greater perturbation in MRSA membranes as compared to the results obtained from the materials tested separately.

A synergy has also been confirmed in CS–guanidine complexes prepared by reacting CS and polyhexamethylene guanidine hydrochloride (CS-PHGH) [38], and in this case, the MIC value of a CS–PHGH complex against *E. coli* was 15.6 µg mL^−1^, whereas the MIC values corresponding to pristine CS and PHGH were 1280.0 and 16.3 µg mL^−1^, respectively. In another study, CS was blended with different nanocrystalline celluloses, which were previously modified with quaternary ammonium salts, such as epoxypropyltrimethylammonium chloride, *N*,*N*-dimethyl-*N*-dodecyl-*N*-(1,2-epoxypropyl) ammonium chloride (DMDEPAC) and *N*,*N*-dimethyl-*N*-octadecyl-*N*-(1,2-epoxypropyl) ammonium chloride (DMOEPAC). The antibacterial tests of these materials revealed good biocidal efficacy against *S. aureus* and *E. coli* O157:H7 and, notably, the systems modified with DMDEPAC demonstrated a 6-log reduction of the concentration of both strains within 5 min [39].

Apart from CS, the combinations of various biocidal functionalities utilizing different polymeric systems have been described. For instance, Xiao et al. modified polyacrylamide (PAM) with both quaternary ammonium salts (QAS) and quaternary phosphonium salts (QPS) moieties [40]. In general, QPS polymeric compounds display higher antimicrobial activity compared to the corresponding polymeric QAS [41], probably because phosphorus atoms favor the adsorption process, that is, positively charged moieties on negatively charged bacterial membranes. In this latter contribution, a triblock copolymer was prepared from acrylamide (AM) monomer and (4-penten-1-yl)-triphenylphosphonium bromide (PTPB) and diallyl dimethyl ammonium chloride (DADMAC) as cationic comonomers. The antimicrobial activity of this material with dual active groups was investigated against *E. coli* bacterium and non-enveloped adenovirus (ADV) demonstrating that the incorporation of a 55% content of PTPB in the copolymer depresses the MIC value to 75 µg mL^−1.^ Therefore, this provides a high viricidal efficiency and suggests a dual-functional antibacterial/antiviral activity.

### 2.2. N-Halamines Based Polymers and Polycations Working in Tandem

In addition to cationic polymers containing QAS, QPS or guanidinium salts, polymers bearing *N*-halamines have been demonstrated to be excellent broad-spectrum biocides, with long-term stability, low toxicity and relatively low cost [42,43,44,45,46,47]. *N*-Halamines refers to any compound containing one or more nitrogen atoms, which are usually in the form of imide, amide or amine groups, and form covalent bonds with halogens (N−X), generally chlorine, but also bromine or iodine (Figure 2) [42,48]. The biocidal mechanism of action of these compounds is mainly via an oxidative process from halogen atoms to thiol and amino groups found in natural protein receptors, resulting in the inhibition or inactivation of cells [19,49,50]. This biocidal function of *N*-halamines can be regenerated via a reaction with halogen-donor compounds, such as sodium hypochlorite or sodium hypobromite.

The mechanism of action of *N*-halamine-based polymers differs from the one observed in other cationic polymers. Therefore, extensive research has been focused on combining *N*-halamine functionality with cationic groups, in particular with QAS, aiming at new antimicrobial systems with enhanced activity and synergistic features. This is of particular interest in fighting Gram-negative microorganisms, which have a double lipid bilayer (inner and outer membrane) and limit the diffusion of biocidal agents. In this matter, Worley and coworkers prepared several antimicrobial systems combining *N*-halamine with QAS. Some of their reports showed that the incorporation of QAS and *N*-chloramine considerably enhances the water solubility of copolymers based on these compounds, but had a small effect on the antibacterial activity [51,52,53]. In contrast, others investigations demonstrated the enhanced biocidal effects of immobilized QAS linked to various *N*-halamine derivatives. For example, Li et al. [54] synthesized several *N*-chloramine precursors containing either QAS or QPS (Figure 3), which were attached to PET or cotton surfaces via click chemistry. The antibacterial activity of these chlorinated systems was tested against MRSA and *E. coli* in solution and on polymer surfaces. These cationic groups showed a positive contribution to the chlorination process and improved the antibacterial action. The proposed mechanism of action for these antibacterial materials is illustrated in Figure 3 and involves electrostatic interactions between the cationic groups of QAS and negatively charged bacterial cells, which facilitate the oxidative chlorine transfer from *N*-chlorohydantoin to the biological receptors followed by bacterial death.

**Table 1 polymers-11-01789-t001:** Antimicrobial investigations of systems containing *N*-halamines and quaternary ammonium salts (QAS) moieties attached to different surfaces.

Schematic Representation of the Polymeric Agent		Surface	Micro-Organism	Percentage Reduction of Bacteria ^1^ (Representative Examples)	Contact Time	Ref.
**HQ_1_**	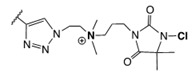	Cotton (C) Poly(ethylene terephthalate) (PET)	MRSA *E. coli*	MRSA: 2.0 × 10^6^ CFU/mL Unmodified-C = 0% HQ_1_-*g*-C = 27.2% Chlorinated H_1_-*g*-C = 100%	5 min	[54]
HQ_2_	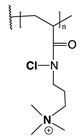	Cotton (C)	*S. aureus* *E. coli*	*S. aureus:* 6.67 × 10^5^ CFU/mL Unmodified-C = N/A HQ_2_-*g*-C = 89.55% Chlorinated X-*g*-C = 100%	5 min	[55]
HQ_2_	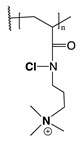	Mesoporous silica SBA-15	*S. aureus* *E. coli*	*E. coli:* 3.3 × 10^7^ CFU/mL SBA-15: 2.27% SAB-15-*g*-HQ_6_ = 24.1% Chlorinated SAB-15-*g*-HQ_6_ = 100%	10 min	[56]
H_1_-Q_1_	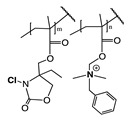	Cellulose fiber (Ce)	*S. aureus* *E. coli*	*E. coli*: 10^6^ to 10^7^ CFU/mL ^a^Unmodified-Ce = 1.0% H_1_-*g*-Ce = 99.0% Q_1_-*g*-Ce = 93.0% Chlorinated H_3_Q_1_-*g*-Ce = 99.5%	5 min	[57]
HQ_3_	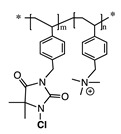	Macroporous crosslinked chloromethylated polystyrene (CMPS) resin	*S. aureus* *E. coli*	*E. coli*: 1.8 × 10^7^ CFU/mL Unmodified-CMPS = 32.39% HQ_3_-*g*-CMPS = 63.69% Chlorinated HQ_3_-*g*-CMPS = 99.99%	5 min	[58]
HQ_4_	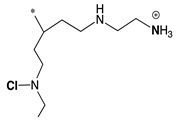	Polypropylene (PP)	*L. mono-cytogenes*	*L. monocytogenes:* 1.0 × 10^6^ CFU/mL ^a^Unmodified-PP ≈ 0% Chlorinated-PP ≈ 0% HQ_4_-*g*-PP ≈ 99.9% Chlorinated HQ_4_-*g*-PP > 99.99%	120 min	[59]
H_2,_ Q_2_	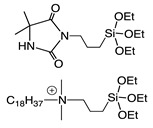	Cotton (C)	*S. aureus* *E. coli*	*E. coli*: 2.51 × 10^6^ CFU/mL C + H_2_ = 2.27% Chlorinated-C + H_2_ = 100% C + Q_2_ = 46.3% Chlorinated C + H_2_ + Q_2_ = 100%	30 min	[60]
HQ_5_	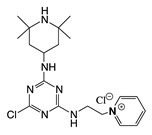	Cotton (C)	*S. aureus* *E. coli*	*E. coli*: 1.93 × 10^6^ CFU/mL Unmodified C = 30.57% C-*g*-HQ_5_ = 51.4% Chlorinated C-*g*-HQ_5_ = 100%	30 min	[61]

^1^ Percentage reduction values were calculated from P= (1 − 10^−L^) × 100, where P is percent reduction and L is log reduction H = *N*-Halamine, Q = cationic compound, HQ = *N*-Halamine and cationic moieties in the same chain.

Separate investigations of antibacterial systems that integrate *N*-chloramines and QAS attached to different surfaces are summarized in Table 1. In general, the reported results suggest that the presence of both functionalities provides a potent antibacterial capacity. However, it seems that the antimicrobial activity of these materials is more pronounced due to the presence of *N*-chloramines, rather than the QAS moieties, suggesting that the immobilization of the QAS moieties onto surfaces hinders their ability to diffuse into cell walls limiting their biocide capability (in particular in Gram-negative bacteria) [62]. Nevertheless, a part from promoting electrostatic interactions with bacteria, cationic charges may also cause cellular damage through an ion exchange process [63]. Generally, dual systems based on *N*-halamines and QAS have excellent antimicrobial properties achieved through two different mechanisms of action.

### 2.3. Nitric Oxide co-Administration Systems

The NO-based polymers containing other biocidal moieties also represent an alternative for the development of more effective antimicrobial materials to fight drug resistance. Over the last years, antimicrobial therapies based on NO demonstrated a robust strategy to kill the most common infectious pathogens [64,65]. The outstanding antimicrobial properties of NO are derived from its ability to induce irreversible cellular damages via oxidative stresses to different cellular components (*v.gr*., DNA, proteins) by using reactive byproducts such as dinitrogen trioxide and peroxynitrite. These processes are schematically depicted in Figure 4 [66]. In addition, the generation of NO resistant strains is limited with this strategy because, apart from an extremely rapid reduction in microbial populations, its mechanism of action is unspecific.

Due to the great potential of NO as an antimicrobial agent, considerable research efforts have been made on the synthesis of NO donor molecules, such as *S*-nitroso-*N* acetylpenicillamine (SNAP), which allow the storage and posterior delivery of NO. Likewise, different strategies have also incorporated this NO donor into polymeric materials [67]. For more than a decade, Schoenfisch’s group has focused on investigating NO-based antimicrobial polymers obtaining promising results and, more recently, on exploring approaches for the combined therapies in attempts to increase bactericidal efficacy for a broader microbial spectrum. In this context, interesting contributions dealing with dual systems that integrate NO-releasing compounds with other biocides have been reported aiming at reducing microbial resistance acting synergistically. These systems include antimicrobial dendrimers [68,69,70,71], quaternary ammonium polymers [72], silver derivatives [73,74] or chitosan [75]. Thus, the combinations of NO releasing compounds with antimicrobial polymers containing different biocidal functionalities are discussed next. 

Recently, a multi-defense strategy for the preparation of a novel antimicrobial system has been described, where a NO releasing agent (*v.gr*., SNAP) and benzophenone based quaternary ammonium molecules (BPAM) [76] were incorporated onto CarboSil^®^ polymer (a poly(silicone–carbonate-urethane) terpolymer used in the fabrication of medical devices). Interestingly, the surfaces containing both agents (SNAP/BPAM) exhibited a significantly enhanced antimicrobial activity against *P. aeruginosa* and *S. aureus* as compared to both agents tested individually (Figure 5). It was proposed that a sustained release of NO acts against bacteria adhered on these treated surfaces, lowers their viability and favors the biocidal action of BPAM by contact.

Boyer et al. [77] incorporated NO donor functional groups into a linear amphiphilic terpolymer formed by biocompatible oligoethylene glycol, hydrophobic ethylhexyl and cationic primary amine segments. A cooperative effect was observed in the biofilm dispersal, planktonic and biofilm killing tests against *P. aeruginosa.* In this specific case, the terpolymer had an analytical performance acting as an antimicrobial agent by promoting bacterial death via a membrane wall disruption, as well as NO carrier.

NO-releasing compounds have also been combined with antimicrobial dendrimers, well-defined macromolecules with highly branched structures that emulate natural architectures, in which their size and number of arms can be tailored [78]. Based on this approach, the Schoenfisch’s group reported that dendrimers can be useful scaffolds for the development of combined therapies [68,69,70,71]. For example, they demonstrated that the addition of NO donors into PAMAM dendrimers modified with QAS of different alkyl chains considerably improves the biocidal performance as compared to non-NO-releasing dendrimers, which were poorly effective to prevent the biofilms formation of *S. aureus* and *P. aeruginosa* [70]. The formation of dangerous bacterial biofilms on the surface of medical devices, such as catheters or implants, often leads to health complications, such as recurrent infections [79]. The treatment of biofilm-related infections is challenging because the bacteria cells are embedded in a polysaccharide matrix with embedded cells which makes them highly resistant to hostile environmental conditions and antibiotic treatment [14].

At this time, the synergy between NO-releasing agents and dendrimers remains uncertain since only a low cooperative effect has been observed. As a consequence, a comprehensive study is required to clarify a possible collaborative interaction, as well as the extent of such an impact. It may also be desirable to design new polymers to combine them with NO-releasing compounds and other biocidal agents. In summary, NO-releasing agents appear to be a promising research area for the development of novel antimicrobial materials to fight bacterial drug resistance.

## 3. Enhancing Efficiency of Antimicrobial Cationic Polymers by Incorporating Metal Oxides and Metal Nanoparticles

Nanomaterials provide attractive structural and surface properties to fight bacterial infections due to their exceptional diffusivity and penetrability, as well as their ability to inhibit bacterial growth. During recent decades, substantial research on their biocide properties has been performed opening their application to several healthcare fields such as nanomedicine, drug delivery or pharmacy [80]. Thus, the use of combinations of nanoparticles (NPs) with cationic antimicrobial polymers represents a promising approach to reduce microbial resistance. In particular, NPs derived from silver, copper, gold, titanium oxide as well as zinc oxide have shown outstanding bactericidal behavior and a wide range of antimicrobial effects, and have arisen as attractive alternative materials to complement and enhance the performance of well-known antimicrobial polymers [81,82,83]. Thus far, the mechanisms of action of NPs against bacteria have not been precisely established and depend on many factors (size, concentration, surface modification, etc.), but it is known that they cause damage on the cell membrane altering vital functions as permeability and respiration, and disturb functions of some proteins contained into bacteria [84]. It is well documented that silver possesses strong antimicrobial properties both in its salt and nanoparticle form. Hence, it has found a variety of applications in different fields [85,86,87].

In particular, silver nanoparticles (AgNPs), one of the predominant nanomaterials, display a broad-spectrum of antimicrobial activity against bacterial and fungal species, including antibiotic-resistant strains [88,89]. Their proposed mechanisms of action involve the gradual release of silver ions that interfere in DNA functions, direct damage to the cell structure and the generation of reactive oxygen species (ROS) [86,90]. In case of copper nanoparticles (CuNPs), their metallic and ionic forms produce toxic hydroxyl radicals that damage essential proteins and DNA [84]. Furthermore, gold nanoparticles (AuNPs) have shown to combat multi-drug-resistant pathogenic bacteria, and also, to exhibit enhanced properties in terms of physical and chemical stability with excellent biocompatibility [91,92]. ZnO and TiO_2_ NPs have also attracted considerable interest due to its good photocatalytic activity, nontoxicity and antibacterial properties for application in several fields such as cosmetics, depollution and protective medical clothes [93]. The commonly accepted mechanism of action of these NPs is the production of ROS, although, in the case of ZnO NPs, the release of Zn^2+^ also contributes to its activity.

Some interesting studies focused on taking advantage of antimicrobial properties of polymers, as well as NPs, applied on surfaces, coatings or fibers have been reported in the last decade. In the pioneering studies, Sen and coworkers published the enhanced and long-lasting antimicrobial properties of a dual system [94]. In this report, spherical silver bromide nanoparticles (AgBr) were embedded into an antimicrobial polymer matrix of poly(4-vinyl-*N*-hexylpyridinium bromide) (NPVP), i.e., a polymer that causes cell death by disrupting the membrane bacteria [95]. The pyridine groups acted as a capping agent stabilizing the AgBr nanoparticles and preventing the formation of agglomerates (see Figure 6). These AgBr nanoparticles are responsible for releasing Ag+ ions.

This takes advantage to cause bacteria annihilation by a combination of mechanisms which include inactivating enzymatic functions due to interactions with thiol groups, [96,97] binding to DNA and disrupting membrane permeability [98]. The effectiveness of the AgBr/NPVP nanocomposite was proven in terms of the low MIC values found against *E. coli* and *B. Cereus* (see Table 2). In this additive effect, two mechanisms of action play an important role: On one hand, the amphiphilic polymer damaging the structure of bacteria by contact and, on the other hand, stable AgBr nanoparticles releasing Ag^+^ biocidal ions allowing the faster elimination of microbial colonies in comparison to the separated effects of the polymer and AgBr nanoparticles alone.

Multiple systems containing different synthetic cationic polymers and metal NPs disclosed in the last decade are summarized in Table 2. The enlisted studies described briefly the mechanism of action to enhance antimicrobial activity. In most cases, positive charges attached to polymers help to bind their chains to the surface of bacteria causing structural damages and increasing permeability allowing the NPs to diffuse quickly into bacteria. In this in-collusion acting process, two mechanisms of action are involved, i.e., contact and release. Adsorption of cationic polymers as well as NPs onto cell may increase the permeability of bacteria facilitating the diffusion of NPs into the interior, prompting the release of ions, disrupting the cell wall, and triggering the ejection of vital components. Furthermore, radicals or oxidative species also attack fundamental components, like DNA and proteins. Likewise, if bacteria exhibit resistance to one of the agents, another one kills the microorganism acting in a different mode.

Binary systems are not only limited to coatings. They can be extrapolated to other systems, such as nanofibers. Likewise, other cationic polymers, not containing QAS moieties, have been also proven as efficient antimicrobial agents. In this regard, Jang and coworkers obtained in one-step AgNPs embedded into poly[2-(*tert*-butylaminoethyl) methacrylate] nanofibers (Ag/PTBAM) [99]. The large pendant secondary amino group contained in PTBAM causes phase separation of the lipid layers inside the bacteria resulting in cell death, and it also possesses low toxicity against human cells [100]. Figure 7 details the synthesis of Ag/PTBAM nanofibers by radical-mediated dispersion polymerization. Silver ions (Ag^+^) were first coordinated with hydroxyl groups of poly(vinyl alcohol) (PVA). Subsequently, the TBAM monomer was polymerized. The antibacterial test revealed that a combination of AgNPs and PTBAM enhanced the antibacterial efficacy against *E. coli* y *S*. *aureus* in comparison to silver nitrate, AgNPs/PMMA nancomposite and the anti-inflammatory silver sulfadiazine (SSD) due to a considerable increase in their surface area which, in turn, leads to more cell attachment and higher silver release.

Therefore, combinations of non-depleting biocides, such as antimicrobial polymers, with a second biocide able to release toxic agents against bacteria, such as nanoparticles, represent promising approaches for multimodal therapeutics which may enlarge their application window.

The combination of biocidal NPs and CS deserves specific attention. This combination, as aforementioned, has demonstrated a broad antimicrobial activity against many microorganisms [25]. It is generally accepted that polycationic chitosan can bind to negatively charged cell membranes leading to a decrease in the osmotic stability of the cell and followed by the subsequent leakage of intracellular constituents [108]. Despite its biodegradability and biocompatibility, in recent years there has been considerable interest in the design of CS/metal NP (i.e., AgNPs, ZnO, CuNPs) materials due to their wide range of applications as biomaterials, wound or burn dressings.

According to Muzzarelli, CS possesses a high ability to bind metal ions via chelation with amino groups [109], a property that has been used for the synthesis of several systems such as coating, fibers, etc., where CS not only acts as antimicrobial agent, but it is essential to achieve a good dispersion of metal NPs. It works as an ion capping agent to control the growth of NPs and avoids their aggregation. Several studies reported the synthesis of composites based on a combination of AgNPs [110,111,112,113,114] and AuNPs [115] with CS, in which the biocidal action is exclusively attributed to nanomaterials, preventing a synergistic effect. This phenomenon may be attributed to CS as it loses its positive charges via a neutralization process or chemical modifications, which limit its antibacterial properties. However, there are interesting contributions related to CS acting in coordination with one or more agents to inactivate a wide range of bacteria. For example, an interesting three-component antibacterial compound was reported by Banerjee and coworkers [108], who investigated the synergy in the bactericidal potency of a CS-AgNPs composite in the presence of molecular iodine. Iodine is a well-documented broad-spectrum agent that inactivates bacteria by affecting functions of enzymes and cell proteins and altering the membrane structure [116]. It was demonstrated that the iodinated composite presented higher antimicrobial activity at a lower dose compared to the effective dose of the individual components. Figure 8 shows the proposed mechanism of action, where the positively charge CS chains attach to the negative cell wall, while the AgNPs dispersed in the matrix turn the cell wall porous and activate the formation of ROS. Furthermore, the AgNPs also produced iodine atoms from iodine molecules deposited on their surface. These iodine atoms induced an enhanced production of ROS, which causes further damage to the cell. Overall, CS, AgNPs and iodine (which, in turn, forms an iodine atom) work in tandem for superior antibacterial activity in comparison to either of the components, and at lower concentrations of each of the species in the composite than their individual potency. This composite had the advantage that the concentration of AgNPs could be minimized. Similarly, this group studied CS-CuNPs composite in the presence of molecular iodine which provided stability to NPs in the media [117]. The bactericidal activity of this composite was determined on *E. coli*, where the MIC was found to be 130.84 µg mL^−1^, which consisted of 127.62 µg mL^−1^ of CS-CuNP composite and 3.22 µg mL^−1^ of iodine. The high effectiveness of this biocidal composite was attributed to polycationic CS which interacts with the negatively charged cell envelope. Furthermore, CuNPs possibly became attached to sulfur groups contained in the proteins of the bacterial cell membrane. This interaction can cause permeability of the membrane leading to leakage of proteins and other intracellular constituents and the death of bacteria. Another investigation regarding ternary antimicrobial systems was performed by Sant’Ana and coworkers who involved CS, AgNPs and some antibiotics [118].

Besides AgNps, in the last decade, recent studies on ZnO/CS composites have demonstrated that the presence of ZnO NPs alone or in combination with other biocidal NPs remarkably enhances the antibacterial properties of CS [119]. For example, Li and coworkers prepared a novel CS/Ag/ZnO blend to prepare films where Ag and ZnO NPs with spherical and granular morphology were uniformly distributed into the CS polymer [120]. The test of antimicrobial activities on a broad range of bacteria, including *B. subtilis*, *E. coli*, *S. aureus*, *Penicillium*, *Aspergillus*, *Rhizopus* and yeast colonies showed that CS/Ag/ZnO films had higher antimicrobial activities than films derived from CS/Ag and CS/ZnO blends, indicating that the composite of Ag and ZnO enhanced the biocidal properties of CS. In the case of ZnO NPs, when these are under light irradiation, electron–hole pairs are generated. The holes (h+) react with OH− on the surface of NPs, creating hydroxyl radicals (OH·), superoxide anion (O_2_^−^) and perhydroxyl radicals (HO_2_·). These highly active free radicals damage the cells of microorganisms as a result of decomposition and ultimately destruction. With the presence of AgNPs, charge transfer was improved, reducing the chance of electron–hole pairs to recombine and promote the generations of perhydroxyl radicals and other active oxidizing materials. Therefore, the presence of Ag and ZnO significantly enhanced the antimicrobial ability of CS.

Additional selected examples of CS-based antimicrobial systems working in combination with biocidal NPs as ZnO, AgNPs, etc., are summarized in Table 3.

Although ferrites have not attracted as much attention as the case of silver and copper derivatives, ferrite superparamagnetic NPs have potentially antibacterial activity [128]. In particular, it has been demonstrated that superparamagnetic MnFe_2_O_4_ NPs cause a leakage of lactatedehydrogenase from the bacterial membrane, mytocondrial function disturbance, chromosomal condensation, and oxygen free radicals’ production. In this regard, Esmaeili et al. prepared a vancomycin-PEG-CS-MnFe_2_O_4_ NPs (vanco-PEG-CS-MnFe_2_O_4_) that showed an enhanced effect against *E. coli* and *P. aeruginosa* in comparison to antibiotics alone. The PEG chains increased the stability of NPs while CS coating not only improved the properties of ferrite nanoparticles, but promoted antibacterial activity managing a reduction in a dose of vancomycin.

Magnetic hyperthermia is a therapeutic procedure that increases temperature in tissues and based on the fact that magnetic nanoparticles (MNPs) that are exposed to an alternating magnetic field absorb energy which efficiently produces localized heat [129]. This strategy has been applied in cancer treatment [130] and recently as an alternative method to inactivate the *S*. *aureus* pathogen [129,131]. A novel dual antimicrobial system that takes advantage of magnetic hyperthermia benefits has been reported by Duan and coworkers [132], who developed hybrid antimicrobial NPs with cationic polycarbonate brushes (PrBrT) grafted on super-paramagnetic MnFe_2_O_4_ NPs (10 nm). The system MnFe_2_O_4_-*g*-PrBrT was evaluated against *E. coli* and *S. aureus* strains, as described before. Cationic species in the polymer chains were responsible for cell disruption and leakage of components. The synergistic effect was observed when magnetic MnFe_2_O_4_-*g*-PrBrT attached to bacteria was exposed to magnetic heating resulting in a localized increase of temperature killing them efficiently.

## 4. Synergistic Effect Between Antimicrobial Polymers and Carbon Nanostructures

Carbon-based nanomaterials (CBNs) have become crucial for a wide range of applications, including biomedical applications [133]. Functionalized CBNs have extensively been used as carriers of antibiotics, which would decrease resistance and enhance their bioavailability. Furthermore, CBNs have also shown potent antimicrobial properties by themselves, in particular carbon nanotubes (CNTs) and graphene oxide (GO) [134].

The features of CNTs attracted considerable attention as nano-reinforcements for advanced polymer nanocomposites in several fields since Iijima first discovered them in 1991 [135]. It is well-known that CNTs have outstanding mechanical, electrical and thermal properties due to their remarkable structure [136]. In addition to the above mentioned features, in recent years it has been reported that CNTs have the ability to eliminate bacteria such as *E. coli*^,^ and *S. aureus* in solution [137,138]. Figure 9 shows the proposed biocidal effect associated to a physical interaction of nanotubes acting as needles on the surface of microorganisms, which causes highly localized degradation of bacterial cell walls [139]. However, other secondary factors may also contribute to bacteria destruction, such as inhibiting cell growth, oxidative stress and the presence of metal residues [140]. Previous results demonstrated that the antibacterial activity of CNTs depends on some physical factors such as length, diameter, dispersion and concentration [137,140,141].

The synergistic performance of CNTs combined with some polycations was explored by Kim and coworkers, who synthesized multiwalled carbon nanotubes (MWCNT) functionalized with poly(2-dimethylaminoethyl methacrylate) (PDMAEMA) [142]. Furthermore, PDMAEMA chains were quaternized with bromoethane and the resulting hybrid material clearly revealed an antibacterial effect against *E. coli* as well as against *S. aureus*. It was shown that the loss of viability of *E. coli* highly depended on the PDMAEMA content on the MWCNT surface. Various more complex cationic polymers, such as dendrimers, have also been used to functionalize MWCNT aiming at enhanced biocidal properties. Murugan and coworkers evaluated the antimicrobial performance of MWCNT modified with an amphiphilic cationic dendrimer and silver nanoparticles [143]. The hybrid nanostructures were synthesized from carboxylated MWCNT (MWCNT-COOH). On the one hand, poly(propylene imine) dendrimer (PPI) were attached to MWCNT-COOH obtaining a MWCNT-PPI nanohybrid. On the other hand, AgNPs were deposited on MWCNT-PPI, yielding a silver-dendrimer complex MWCNT-PPI-Ag. The results demonstrated that the antimicrobial activity of both nanohybrids against *B. subtilis*, *S. aureus*, and *E. coli* was achieved following the order of efficiency in terms of percentages of the kill of MWCNTs-APPI-AgNPs > MWCNTS-PPI > MWCNTS-COOH. As in previous cases, quaternary ammonium groups helped to increase permeability of the cell membrane, and therefore, significantly enhancing the antimicrobial activity against the *B. subtilis*. In the case of MWCNTs-PPI-AgNPs (Figure 10), the cooperative effects of AgNPs deposition onto MWCNTs-PPI lead to an improvement of the action against the microorganisms and, hence, it was able to destroy *B. subtilis* and *S. aureus* bacteria more effectively than the other two materials.

Graphene is a single atomic plane of hexagonally arranged carbon atoms obtained through the exfoliation of tridimensional graphite [144]. The sp^2^ hybrid carbon framework confers to graphene materials (GMs) excellent thermal, mechanical and electrical properties which are applied to the design of nanoelectronics, biosensors and transistors [145]. Several studies have demonstrated the high antimicrobial activity of graphene and graphene oxide derivatives against a wide variety of microorganisms, including Gram-positive and Gram-negative bacteria, with little resistance and a tolerable effect on mammalian cells [146,147,148,149]. Although the mechanism of action of graphene is not entirely elucidated, in general, as in the case of CNTs, it involves both physical and chemical effects where nanosheets are in direct contact with microorganisms [150]. Physical results mainly imply bacterial membrane damage caused by the sharp edges of graphene and the subsequent leakage of intracellular substances, which ultimately leads to bacterial death. Furthermore, the wrapping or trapping mechanisms might be also involved, avoiding bacteria proliferation by reducing the microbial metabolic activity. On the other hand, the chemical effect is principally associated to a charge transfer mechanism or to an oxidative stress mediated by ROS.

Modern approaches combining GMs with other biocidal agents as quaternary ammonium [151] and phosphonium salts [152], silver [153] as well as peptides [154] have been successfully used to develop water disinfection membranes, hydrogels or coatings with enhanced properties, which can be applied in biological and medical sciences. Besides, it has been demonstrated that the antimicrobial performance of GMs improves when combined with cationic polymers. For example, a nanocomposite from GO and CS with enhanced antibacterial properties was reported by Harikarthick et al. [155], where an increase in the roughness of CS-GO surface (in comparison to the smooth morphology exhibited by GO) facilitated bacterial cell adhesion, which finally caused the death of a higher number of *B. subtilis* and *E. coli* cells.

Therefore, CNTs and GMs represent an excellent alternative to generate active dual antibacterial compounds due to the aforementioned intrinsic properties. Their antimicrobial nature strongly depends on the size and dispersion to get contact between the CNTs/GMs and the bacterial cell surface. Nevertheless, the major drawback of these carbon structures is that they tend to self-associate and aggregate owing to strong van der Waals interactions between them, which would prevent direct contact with the bacterium causing a decrease in antibacterial activity. Thus, it is necessary to decrease this physical interaction with an efficient exfoliation process and/or via surface functionalization.

## 5. Combination of Antimicrobial Cationic Polymers with Antibiotics

Although nowadays there are many efforts in developing new effective antibiotics, the number of such drugs that are available in the market is somewhat limited due to the rising costs for the development and approval of medications [156]. The derivatization of existing antibiotics and their combinations represent promising alternatives to shorten the time of expansion for new antimicrobial agents. Additionally, to further improve the antimicrobial activities of antibiotics, considerable research has been focused on complex or chemically attach these drugs to biocidal polymers through their functional groups. By this co-administration, it is possible to take advantage of antimicrobial properties of polymers in combination with the lethal action of antibiotics that involves the inhibition of DNA replication [157], among other mechanisms. The application of these dual systems results in preventing bacterial infection with a reduction of the dose of the drug which acquires relevance due to the antibiotic dissemination in the environment that could lead to the development of bacterial resistance.

One of the first attempts to achieve this additive effect was reported by Decker et al. who combined the bioadhesive properties of CS derivatives with the antibacterial activity of chlorhexidine showing an increase in the antiplaque effect [158]. Concerning this chitin derivative, other studies have demonstrated that the combination of antimicrobial CS derivatives with different antibiotics such as tobramycin, clarithromycin and sulfamethoxazole result in a notable synergistic performance against *P. aeruginosa* [159,160].

Likewise, CS, and other cationic polymers have been combined with antibiotics. For example, He et al. [161] reported the copolymerization of a methacrylate monomer containing ciprofloxacin with a cationic monomer containing protonated primary amine and hydrophobic methyl acrylate to prepare a series of water-soluble amphiphilic statistical copolymers. The evaluation against E. coli showed that a low antibiotic content (4.1% mol) was sufficient to enhance the antimicrobial properties of the copolymer as confirmed by the decrease of MIC values from 40 to 10 µg mL^−1^. This group reported a similar synergic behavior with the incorporation of ciprofloxacin into the backbone of an amphiphilic copolymer based on QAS and butyl acrylate [162]. In both cases, the deactivation of bacteria was the result of the cell disruption caused by the cationic and hydrophobic moieties of the copolymer, plus the inhibition of the activity of the bacterial enzyme DNA gyrase induced by ciprofloxacin. In this context, cationic polyacrylamide has also been employed to potentiate the action of daptomycin against S. aureus biofilms, improving their antimicrobial performance due to the formation of electrostatic interactions between cationic polymers and antibiotics that increase drug accumulation in the bacterial cell [163].

Besides displaying antimicrobial activity, dendrimers can be considered as agents that enhance the therapeutic effectiveness of existing antibiotics due to their well-defined globular structure and their ability to attach drugs onto their surface. The attachment of antibiotics in the dendrimers surface presents some advantages as they may enhance drug solubility or increase their time of release as well as effectively helping to deliver antibiotics. Likewise, the modification of dendrimers with non-toxic derivatives such as maltose or PEG chains may decrease their toxicity to eukaryotic cells. In this regard, Lisowska’s group released some contributions related to the conjugation of commonly used antibiotics such as nadifloxacin [164], ciprofloxacin [164] and amoxicillin [165] with modified poly(propylene imine) (PPI) dendrimers. In these studies, the simultaneous administration of both agents significantly displayed higher antimicrobial efficacy against *S. aureus*, *E. coli* and *P. aeruginosa* in comparison to a free drug allowing for a reduction of the antibiotic dose. Various dual systems based on PEGylated PPI and ciprofloxacin have been designed and tested against *S. aureus* and *C. pneumonia* [166]. Mishra et al. [167] conjugated azithromycin with PAMAM dendrimers for the treatment of chlamydia which demonstrated that this dual agent exhibited an additive effect in comparison to antibiotics alone. In another attempt to prove a synergistic effect between dendrimers and antibiotics, Khalil et al. [168] combined poly(ethylene imine) (PEI) with 10 different families of hydrophilic and hydrophobic antibiotics to test the effect against *P. aeruginosa.* The enhanced antibacterial activity was confirmed for novobiocin, cephalosporins, rifamycins and chloramphenicol, while other antibiotic families, such as polymyxins and fluoroquinolones, showed antagonism and indifference, respectively.

Recently, it has been investigated that ionically charged metallopolymers based on cationic cobaltocenium moieties show antimicrobial activity by reducing β-lactamase enzymes activity and disrupting cell walls. Furthermore, these organometallic compounds can complex antibiotics by the formation of a stable ion-pairing as reported by Zhang et al. [169], who bioconjugated some β-lactam antibiotics, including penicillin-G, amoxicillin, ampicillin and cefazolin to cationic cobaltocenium-containing polymers. Based on the synergistic effect produced by both compounds, this system revealed high efficiency against MRSA with low cytotoxicity attributed to the adsorption of metallopolymer to the negatively charged MRSA surface (similar to other cationic antimicrobial polymers) which promotes damage in the cell walls and at the same time, allows the release of complexed antibiotics (Figure 11).

Concerning the combinations of metallopolymers with antibiotics, Yang et al. [170] proposed a robust three component system based on cationic cobaltocenium, phenylboronic acid and β-lactams together in a single macromolecule. The synergistic effect emerged from the electrostatic absorption to negatively charged cell membranes by the cationic cobaltocenium moiety, while boronic acid attached to lipopolysaccharides on the bacterial cells promoted the reinstatement of the vitality of antibiotics.

Likewise, new antimicrobial systems have been proposed for preventing biofilm formation using existing clinically used drugs attached to a NO donor group. In this respect, Nguyen et al. [171] reported an attractive approach based on polymeric NPs capable of storing NO and gentamicin. The NPs were found to simultaneously release both agents and demonstrated synergistic effects, suppressing the growth of *P. aeruginosa* biofilm and planktonic cultures by more than 90% and 95%, respectively. Furthermore, a novel technology for simultaneously releasing NO and antibiotics (i.e., gentamicin, tetracycline) through an electrochemical NO release catheter device demonstrated a synergistic effect against biofilm and biofilm released cells [172].

Selected examples of antimicrobial polymers co-administrated with antibiotics and its mechanism of action are summarized in Table 4.

## 6. Conclusions

Many strategies have been developed for controlling the proliferation of pathogens. However, there is a critical need to release novel practical approaches such as combined therapies to stop the emergence of resistant species. The concerted use of two or more biocides that act simultaneously decreases the opportunities of life of the pathogenic microorganisms as well as their proliferation by promoting further mechanisms for microbe killing. This review shows the relevance of combining systems based on antimicrobial polymers that have been acquired in the last years. The synergy effect is demonstrated in many of the antimicrobial systems mentioned in this review when two agents working in tandem offer a greater than additive effect, resulting in combinations that are more potent than equivalent doses administered individually, which may reduce potential toxicity to mammalian cells and the cost of treatment. In most of the cases, antimicrobial polymers interact with the bacterial membrane provoking damage and favoring the action of the other biocidal. It was also noted that adding a third component typically increases the magnitude of the antimicrobial activity. Even if combinations do not provide enhanced or synergistic actions, they can broaden the antimicrobial spectrum and reduce the risk of initial inadequate treatments, which are often associated with increased mortality.

In summary, polymers combined with other bioactive substances, synergistically acting in tandem, in antibacterial systems represent an important branch of medical research with broad perspectives. Nonetheless, how to improve the performance of these systems to achieve a level of practical application is our continuous pursuit. Perhaps, the combination with biopolymers or naturally occurring substances may provide dual systems acting in tandem with the potential for accessing a new generation of antibacterial polymers possessing a low health risk. Hopefully, our brief survey is helpful in understanding this strategy and will stimulate some new ideas for future developments.

## Figures and Tables

**Figure 1 polymers-11-01789-f001:**
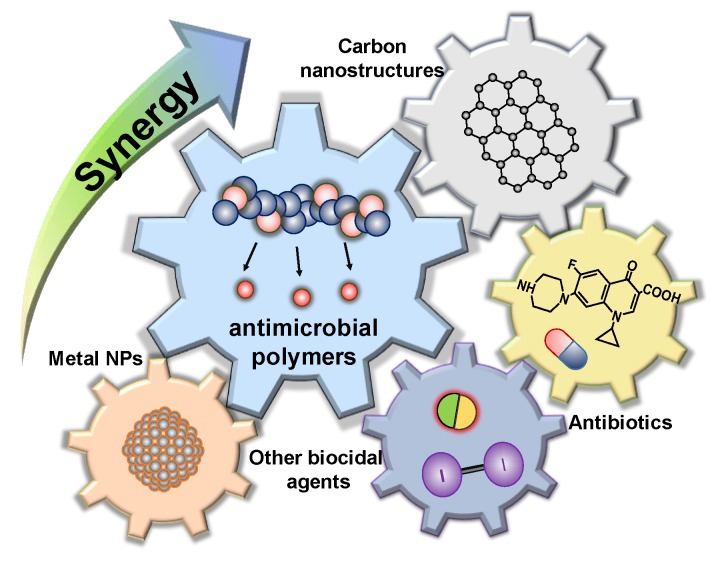
Schematic representation of multicomponent systems based on antimicrobial polymers for enhanced biocidal therapies.

**Figure 2 polymers-11-01789-f002:**
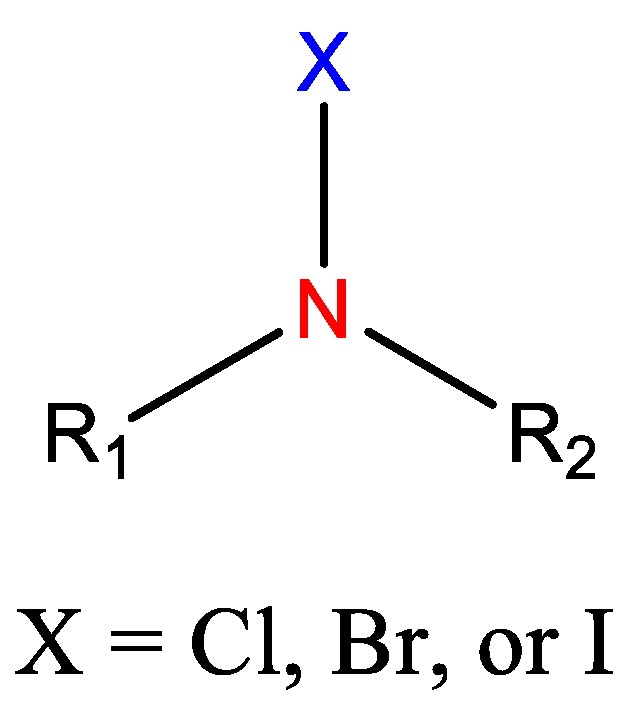
Schematic representation of the general structure of *N*-halamine compounds.

**Figure 3 polymers-11-01789-f003:**
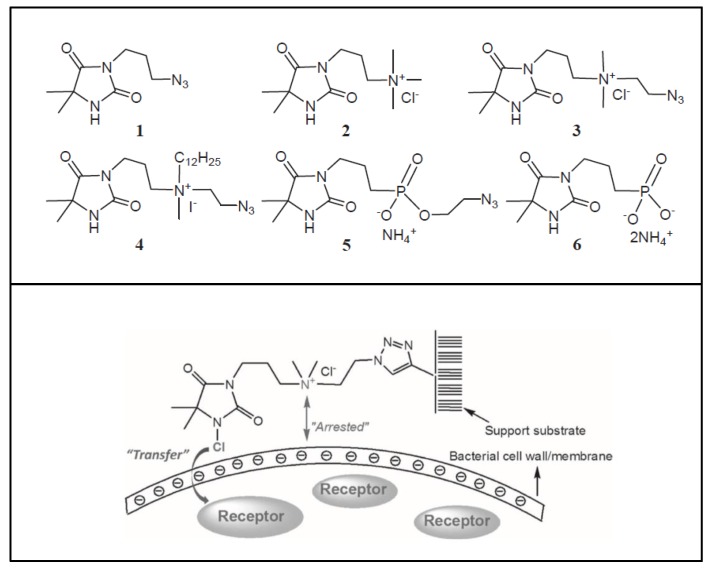
Schematic representations of several *N*-chloramine precursors containing either a quaternary ammonium moiety or a phosphonate functional group (**top**), and of an enhanced antimicrobial function between cationic groups and *N*-chloramine (**bottom**). Reproduced with permission from reference [54]. Copyright 2012 Wiley.

**Figure 4 polymers-11-01789-f004:**
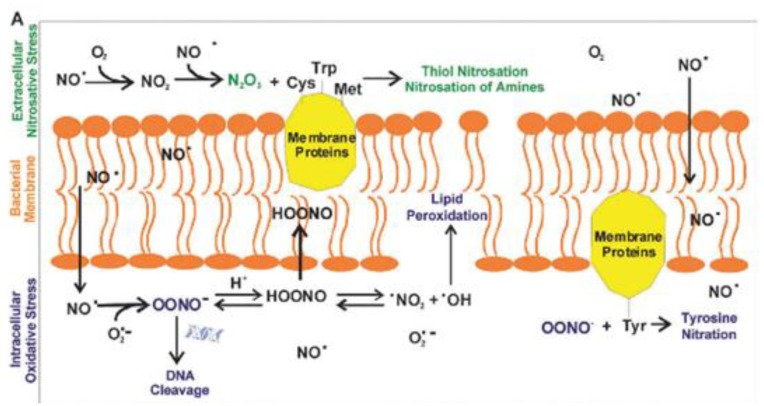
Antimicrobial mechanisms of NO and its byproducts. Reproduced with permission from reference [66]. Copyright 2012 RSC.

**Figure 5 polymers-11-01789-f005:**
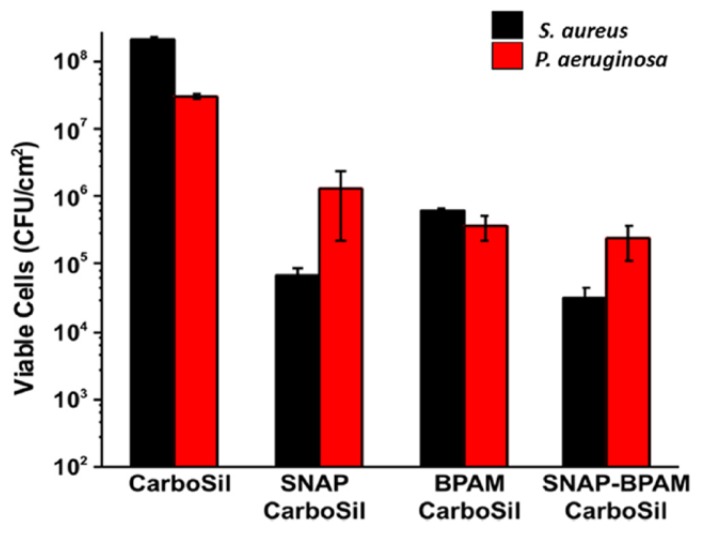
Comparison of the inhibition of viable colony forming units (CFU cm-2) of *S. aureus* and *P. aeruginosa* on the surface Carbosil of *S*-nitroso-*N* acetylpenicillamine (SNAP) films, benzophenone based quaternary ammonium molecules (BPAM) films and SNAP-BPAM films. Reproduced with permission from reference [76]. Copyright 2017 RSC.

**Figure 6 polymers-11-01789-f006:**
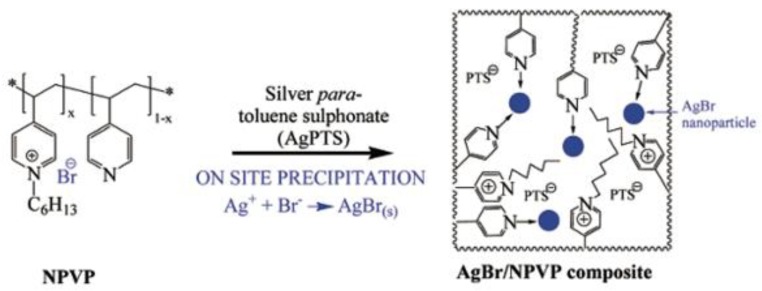
Schematic representation of the preparation of an antibacterial composite material, AgBr/NPVP (NPVP) poly(4-vinylpyridine)-co-poly(4-vinyl-*N*-hexylpyridinium bromide). Reproduced with permission from reference [94]. Copyright 2006 ACS.

**Figure 7 polymers-11-01789-f007:**
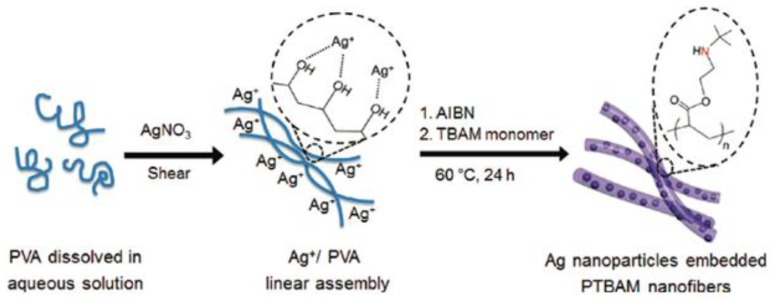
Illustration of the synthetic procedure of AgNPs embedded PTBAM nanofibers. Reproduced with permission from reference [99]. Copyright 2011 ACS.

**Figure 8 polymers-11-01789-f008:**
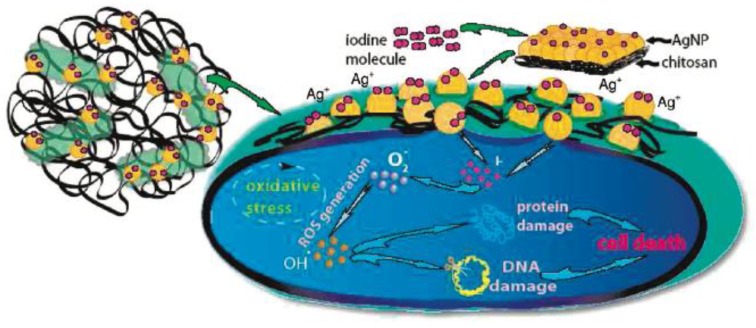
Schematic representation of the proposed mechanism of antibacterial activity of the iodinated CS-AgNPs composite. Reproduced with permission from reference [108]. Copyright 2010 ACS.

**Figure 9 polymers-11-01789-f009:**
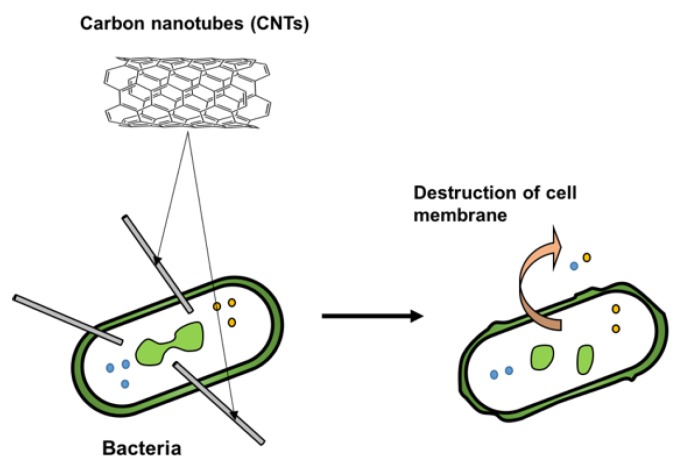
Proposed mechanism of action of CNTs against bacteria.

**Figure 10 polymers-11-01789-f010:**
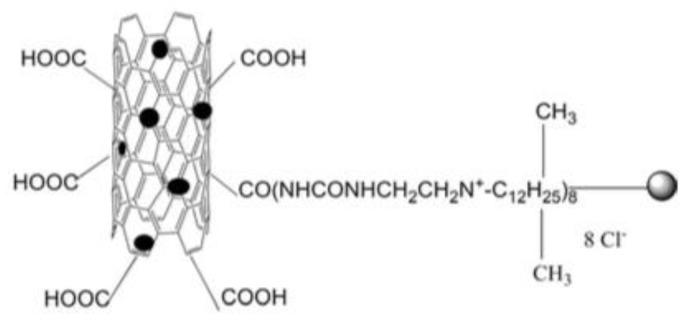
MWCNTs-PPI-AgNP hybrid. Reproduced with permission from reference [143]. Copyright 2011 Elsevier.

**Figure 11 polymers-11-01789-f011:**
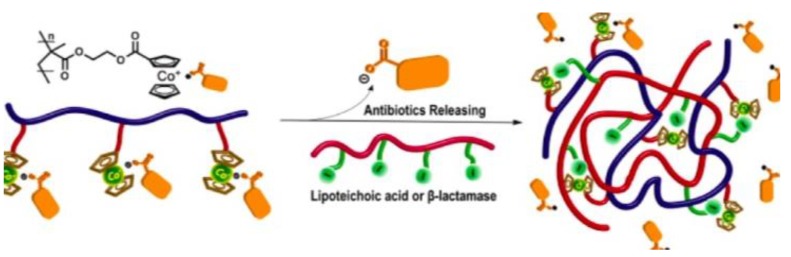
Antibiotic release from antibiotic-metallopolymer ion-pairs via lipoteichoic acid or β-lactamases. Reproduced with permission from reference [169]. Copyright 2014 ACS.

**Table 2 polymers-11-01789-t002:** Mechanisms of action of dual antimicrobial systems based in synthetic cationic polymers and metallic nanoparticles (NPs).

Schematic Representation of the Antimicrobial Polymer	NP/Salt	Microorganism Tested	Synergistic Effect	Ref
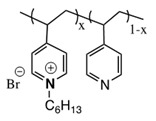	AgBr (10–70 nm)	*E. coli* *B. cereus*	Membrane disrupting of the cationic polymer. Long lasting action without depletion of Ag^+^ ion. The dual system	[94,101]
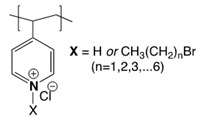	Ag, Cu	*S. aureus* *P. aeruginosa* *B. subtilis* *E. coli*	increased the killing rate of bacteria and kept activity for a longer time in comparison with AgBr alone.	[82]
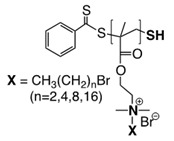	Ag (≈5 nm) Au (≈8 nm)	*S. aureus* *P. aeruginosa*	Positive charges and alkyl chains act together to damage the bacterial structure. This fact increases cell permeability allowing AgNps to penetrate and inhibit the function of enzymes and proteins.	[83,88]
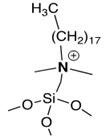	Ag	*E. coli* *S. epidermis*	Release (Ag^+^) and contact killing mechanisms (QAS). Long sustainability.	[102]
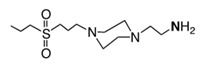	Ag (8 to 15 nm)	*A. niger*	Higher branched degree of polymers produces smaller AgNps with better diffusion and interaction, increasing the antimicrobial performance.	[103]
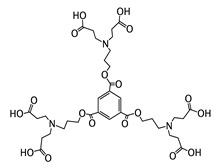	Ag salts	*S. aureus* *MRSA*	Dendrimer acted as a template to load silver salts allowing the high local concentration of exposed silver ions in the periphery.	[104]
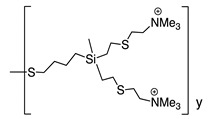	Ag (1.5 nm)	*S. aureus* *E. coli* *P. aeruginosa* *S. hemolyticus* *C. albicans*	Peripheral ^+^NMe_3_ groups in combination with biocidal silver cations.	[105]
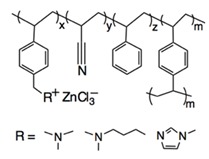	Zn ions	*S. aureus* *E coli* *C. albicans*	Interactions between cations of poly(ionic liquid) and cell wall, which boost the cell membrane permeability causing lysis of the cells. Zn^2+^ can produce reactive ROS in cells leading to the growth inhibition and death of bacteria.	[106]
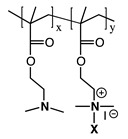	Mg(OH)_2_, Ca(OH)_2_ (30 nm)	*A.niger* *P. oxalicum*	Cationic copolymer provides additional charges on the NPs surfaces promoting affinity to bind to fungal cells, thus improving their interaction with the negatively charged microbial cell surface.	[107]

**Table 3 polymers-11-01789-t003:** Supportive antimicrobial systems based in chitosan and biocidal NPs.

System	Nanoparticle	Microorganism Tested	Synergistic Effect	Ref
Porous CS films	Ag (≈12 nm)	*E. coli*, *S. aureus* *P. aeruginosa*, *MRSA*	The presence of hundreds of porous enables formation of smaller AgNPs, which are more effective than longer. Besides CS absorbs a large amount of water and releases Ag more efficient than chitosan without porous.	[121]
Carboxymethyl CS/polyethylene oxide nanofibers (CMCTS/PEO)	Ag (12 to 18 nm)	*E. coli*, *S. aureus* *P. aeruginosa*, *C. albicans*	The fibrous structure of nanofibers allowed to increase the silver load.	[122]
Crosslinked CS/polyethylene glycol nanocomposite films	ZnO Ag < 100 nm	*E. coli*, *S. aureus* *P. aeruginosa*, *B. subtilis*	Membrane disrupting of the cationic polymer. Ag and ZnO enhanced antibacterial property due to the photocatalysis and metal release process. Generation of active free radicals.	[123]
Nancomposite GO-CS/ZnO	GO ZnO	*E. coli*, *S. aureus*	GO-ZnO induce ROS production that causes oxidative damage. The interaction bacteria with composite and ZnO-NPs increase its permeability and generate active superoxide ions (O_2_^−^), which can react with the peptide linkages in the cell wall of bacteria and thus disrupt.	[124]
CS coatings applied on cotton and cotton/polyester CS/Ag, CS/ZnO, CS/Ag-ZnO	Ag (3 to 5 nm) ZnO Ag-doped ZnO (10 to 35 nm)	*E. coli*, *S. aureus*	AgNPs disturbs the permeability, respiration and cell division. ZnO NPs produce ROS. Under light conditions, Ag improved the charge transfer, reducing the chance of electron–hole pairs to recombine and promoting the generation of perhydroxyl radicals and other potent oxidizing radicals.	[125]
CS NPs	Cu, TiO_2_ ≈10 nm	*E. coli*, *S. aureus*	Negatively charged TiO_2_ NPs acts as a copper ion carrier, and its surface can absorb positively charged copper ions. Cu in combination with TiO_2_ can increase the amount of copper in bacteria and subsequently enhances antimicrobial activity.	[126]
Quaternized CS-clay (MMT) based nanocomposites	Ag (≈26 nm)	*E. coli*, *S. aureus*, *P. aeruginosa*, *B. subtilis*	Exfoliated MMT with a large specific surface area adsorbs and fixes microorganisms. QAS disrupt cell membrane allowing AgNPs infiltrate and react with compounds in the cell wall.	[127]

**Table 4 polymers-11-01789-t004:** Selected antimicrobial systems based in cationic polymers and antibiotics.

Schematic Representation of the Antimicrobial Polymer	Antibiotic	Microorganism Tested	Synergistic Effect	Ref
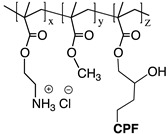	Ciprofloxacin (CPF)	*E. coli*	Integrity of the cell membrane was disrupted by hydrophobic moieties (in an optimal concentration). CPF inhibits the activity of the bacterial DNA gyrase, which leads to bacterial cell death.	[161]
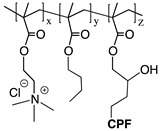	Ciprofloxacin (CPF)	*E. coli*		[162]
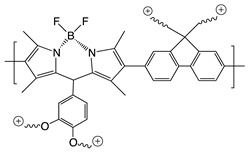	Polypeptide antibiotics: Polymyxin B Polymyxin R	*E. coli*	Combination of cationic conjugated polymers (CCPs) with polypeptide antibiotics facilitates and accelerates the rupture and collapse of bacterial membranes.	[173]
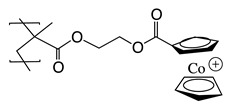	Penicillin-G Amoxicillin Ampicillin Cefazolin	MRSA	Adsorption of metallopolymer to the negatively charged MRSA surface which promotes damage in the cell walls and at the same time allows the release of complexed antibiotic.	[169]
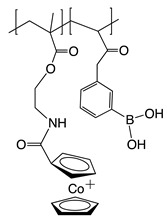	Penicillin-G	*E. coli* *P. aeruginosa* *P. vulgaris*	Phenylboronic acid binds to peptide-glycan via boron-polyol based boronolectin chemistry, cationic cobalto-cenium moiety interact with negatively charged cell membranes and antibiotic is reinstated with enhanced vitality to attack bacteria	[170]

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
