# Peer review of "Combinations of Antimicrobial Polymers with Nanomaterials and Bioactives to Improve Biocidal Therapies"

_polymers, 2019, doi:10.3390/polym11111789_

Round 1
Reviewer 1 Report
A worthwhile contribution.
Author Response
no comments provided by reviewer 1.
Reviewer 2 Report
This manuscript discuss/review the antimicrobial activity of polymers and their combination with other nanomaterials or molecules. Although, this is a relevant subject I only recommend publication after the authors address the following issues:
# 1: The authors should re-write the following paragraph:
“These agents can fight microorganisms by various mechanisms of action ranging from the disruption of cell walls in bacteria to the delivery of metal ions or toxic radicals that inhibit oxidative enzymes or interfere with DNA/RNA replication”
#2: “(…)biocidal activity in the treatment of bacterial infections when they are separately administered, it has been demonstrated that the co-administration of two or more biocides with different mechanisms of action may reduce bacterial resistance which suggests the occurrence of a synergic work”. This sentence must be clarified. Is this a case of reducing resistance or increasing antimicrobial activity?
#3.1: “In general terms, cationic polymers are highly efficient in fighting Gram- positive bacteria by ensuring electrostatic interactions due to their negatively charged cell walls”.
In fact, cationic polymers (and similar to antimicrobial peptides) are also highly efficient against gram-negative bacteria.
#4: “Several reviews summarize different investigations of combinations of biocidal agents for antimicrobial therapies. For example, Allahverdiyev and co-workers address the interactions, mechanisms of action and synergistic effects of different metallic nanoparticle systems (…)”
There are several important works in the field that I clear think the authors should include more relevant references in this manuscript instead of referring to other reviews
I have the same criticism for other sentences along the manuscript
#5: All compounds are in this manuscript identified/divided in sub-topics (e.g. 2.1 Chitosan) with the exception of dendrimers class .
#6: The authors should also explain more clearly the importance/relevance of study the synergy effects of polymers with other biocides. This sentence “To the best of our knowledge, it lacks a comprehensive review of investigations devoted to antimicrobial polymers combined or co-administrated with other biocides to enhance their effectiveness” lacks information.
#7: “Moreover, antimicrobial peptides, e.g., ε-poly-L-lysine (EPL), have also been grafted onto CS chains” Please substitute antimicrobial peptides by antimicrobial polymers or antimicrobial peptide mimics
#8 Table 1: “Antimicrobial investigations of systems containing N-halamines and QAS moieties attached to different surfaces”. The authors include in this table different microorganisms and apparently different S. aureus strains. Apart from MRSA all other S. aureus strains are MSSA?
#9: “(…) improves the biocidal performance considerably as compared to non-NO-releasing dendrimers, which were poorly effective to prevent the biofilms formation of S. Aureus and P. Aeruginosa”. The authors should explain the importance of having a therapeutic treatment that is effective against bacterial biofilms.
#10: “This is of particular interest in fighting Gram- negative microorganisms, which have double cell walls and limit diffusion of biocidal agents”. Please replace double cell wall by double lipid bilayer (inner and outer membrane).
Author Response
in the attached file we respond point-by point o reviewer 2.

Reviewer 3 Report
The present review focused on the combinations of polymers with nanomaterials and bio-actives such as metal oxide, metal nanoparticles, carbon-based nanomaterials, antibiotics, etc. to improve biocidal therapies. The manuscript contains some minor grammar and spelling errors. (1) Authors should write the full name of abbreviations wherever used first time in the review such as v.gr. in line number 70, MIC in line number 114, S. aureus and B. subtilis in line number 116 and so on. Also, full names and their abbreviations are used randomly thought the manuscript. (2) Section 2.1. subtitled as chitosan, but also included the information about N-halamines without any link to chitosan. Authors have explained briefly about bactericidal properties of N-halamines systems which is separate from chitosan. Separate subsection about N-halamines systems and its related table could be made. (3) The review entitled “combinations of polymers with nanomaterials and bio-actives to improve biocidal therapies” but titles of section 2, subsection 2.1 and section 3 are confusing. Similarly, tittle of section 5. I would suggest authors reframe and reorganize all the titles of sections and subsections for better understanding to the readers. (4) Information through tables is a good way to summarize the literature, it should be properly organized for all the sections. Literature is summarized via table only for some sections. It would be better to organize and summarize the table of each section in order to get the quick synopsis of this comprehensive literature. (5) Minor grammar and spelling check is required throughout the manuscript. Such as, reconsider line 256-257, 303-305, 353-354; grammar in line 186, 541; and spelling mistake in line 269, 361 and so on. (6) The literature lacks a description of recent studies. Some literature (especially 2018-2019) along with citation could be added. (7) References contain some errors. Check all the references for errors.
Author Response
see attached file. minor grammar and spelling was done

Round 2
Reviewer 2 Report
The authors improved their manuscript.
Reviewer 3 Report
The authors have answered all the questions. However, several figures in the manuscript are of low resolution, need to be changed.